# Functional Nanoparticles for Enhanced Cancer Therapy

**DOI:** 10.3390/pharmaceutics14081682

**Published:** 2022-08-12

**Authors:** Chenchen Li, Yuqing Li, Guangzhi Li, Song Wu

**Affiliations:** 1Institute of Urology, The Affiliated Luohu Hospital of Shenzhen University, Shenzhen University, Shenzhen 518000, China; 2Department of Urology, South China Hospital, Health Science Center, Shenzhen University, Shenzhen 518116, China

**Keywords:** functional nanoparticles, cancer therapy, nanotechnology

## Abstract

Cancer is the leading cause of death in people worldwide. The conventional therapeutic approach is mainly based on chemotherapy, which has a series of side effects. Compared with traditional chemotherapy drugs, nanoparticle-based delivery of anti-cancer drugs possesses a few attractive features. The application of nanotechnology in an interdisciplinary manner in the biomedical field has led to functional nanoparticles achieving much progress in cancer therapy. Nanoparticles have been involved in the diagnosis and targeted and personalized treatment of cancer. For example, different nano-drug strategies, including endogenous and exogenous stimuli-responsive, surface conjugation, and macromolecular encapsulation for nano-drug systems, have successfully prevented tumor procession. The future for functional nanoparticles is bright and promising due to the fast development of nanotechnology. However, there are still some challenges and limitations that need to be considered. Based on the above contents, the present article analyzes the progress in developing functional nanoparticles in cancer therapy. Research gaps and promising strategies for the clinical application are discussed.

## 1. Introduction

Cancer has a high incidence and mortality rate worldwide [1]. The overall mortality rate of cancer is still as high as 20.2% [2]. Moreover, 19.3 million new cancer cases will occur annually by 2025 [3]. Treatment strategies for cancer mainly depend on the cancer type and the stage of the first diagnosis. Available treatment options for cancer include surgery, radiotherapy, chemotherapy, hormone therapy, immunotherapy, and gene therapy [4]. Among these options, the most common way to treat cancer and inhibit tumor recurrence is chemotherapy, which kills cancer cells with cytotoxic drugs. However, an adverse problem of most chemotherapeutic drugs is that they cannot target the cytotoxicity of tumor cells, resulting in multiple side effects and poor prognosis [5].

In 1986, two Japanese researchers first reported the enhanced permeability and retention (EPR) effect of nanoparticles in tumor tissues, which opened the door to the nano-drug strategy for cancer treatment [6,7]. A large number of subsequent studies showed that, compared with traditional chemotherapy drugs, the drug system based on a nano platform showed significant advantages, such as (1) adjusting the oil–water distribution index of drugs and improving bioavailability, (2) better stability of protein and peptide drugs, (3) targeted administration, (4) the release of drugs in precise doses and on demand, and (5) co-delivery of multiple drugs/diagnostic agents [8,9,10,11]. At the same time, encapsulating anti-cancer drugs has many benefits, including better biological distribution, solubility, and bioavailability [12].

In the present review, we summarize the different functional nano-drug strategies of stimulus-response, surface binding, and macromolecular encapsulation for cancer treatment, discuss the key needs of functional nano-drugs in cancer treatment, and clarify the further prospects of cancer treatment based on nano-drugs (Figure 1).

The main functional nanoparticle types are illustrated here, including tumor microenvironment (TME)-responsive, external stimuli-responsive, surface conjugates, and large molecules.

## 2. Functional Nano-Drug Delivery Platform

Nanotechnology has aroused great interest in cancer treatment because of its excellent solubility, targeting capability, therapeutic efficacy, and low toxicity compared with conventional agents [10]. A series of strategies focused on stimuli-responsive, surface conjugation with targeting ligands, large molecules, and so on offer attractive features and promote highly efficient use of cancerous therapeutic agents. 

### 2.1. Stimuli-Responsive Nano-Drug Delivery Platform

#### 2.1.1. pH-Responsive

Acidic TME caused by hypoxia and extracellular lactic acid accumulation is one of the most significant characteristics of solid tumors. In addition, different cell parts also show different pH values, in which mitochondria are alkaline (pH ~8.0), and lysosomes are acidic organelles (pH ~4.7). During drug delivery, the pH value of endosomes was observed to change from pH ~6.3 to 5.5 [13,14].

A kind of macrophage-membrane-coated nanoparticle (cskc-PPiP/Paclitaxel @Ma) was developed to release tumor-targeted chemotherapy drugs responding to the pH value of endosomes and showed an excellent therapeutic efficacy [15]. One study showed that green-emitting Zn_2_ GeO_4_: Mn^2+^ Pr^3+^ nanoparticles possessed good pH stimuli-responsive luminescent behavior [16]. In a recent study, doxorubicin-hydrazone bond-PEG-folic acid (DOX-hyd-PEG-FA) polymers coated on the surface of nano-graphene oxide could be decomposed at the same time and had good pH sensitivity and active tumor targeting [17]. 

One study revealed that a higher therapeutic effect was seen for melanoma cancer cells than non-pH responsive gold nanoparticles [18]. In a previous study, hydrophobic Curcumin (CUR) was combined with hydrophilic hyaluronic acid (HA) to form pH-responsive nanoparticles, which achieved enhanced treatment efficacy for cancer with good biosafety. The nanoparticle size was 89 nm, and the transmission electron microscope (TEM) image revealed that the morphologies of the nanoparticle were spherical, and the release rate of CUR was 73.5% at pH 5.0 [19]. Furthermore, chitosan, folic acid, and silver nanoparticles loaded with gemcitabine were prepared and had an excellent response to pH. A recent study also demonstrated the responsiveness of layered hydroxide (LDH) in the treatment of colorectal cancer. Under the slightly acidic condition of the tumor site, LDH nano tablets gradually degrade so that Ethylenediaminetetraacetic acid (EDTA) can realize the controllable release of acid reaction at the tumor site. This treatment strategy based on tumor cell separation provides a safe and effective means to treat low-level colorectal cancer and will bring a new dawn to patients with low-level colorectal cancer (Figure 2) [20].

#### 2.1.2. Redox-Responsive

It has been shown that tumor extracellular matrix is an oxidizing medium, while intracellular space is reduced, glutathione (GSH) are reductants widely present in the human body [21,22]. Since the expression of GSH in cancer tissues is higher than that in normal tissues, nanoparticles sensitive to GSH are worthwhile for cancer therapy (Figure 3) [22,23,24]. GSH-responsive hydrophilic PEG and hydrophobic poly (lactic acid co glycolic acid) (PLGA) copolymer were reported to improve therapy efficacy in lung cancer in vitro/in vivo. The nanoparticles were spherical with a diameter of around 200 nm and negative zeta potential [25]. Redox-responsive PEG with PTX NPs achieved a better treatment effect than free drugs in a breast cancer xenograft mouse model. TEM images revealed that the nanoparticles were spherical with an average size of 70 nm [26]. In a previous study, a cationic supramolecular polymer forming a redox-induced gene transfer vector through a host–guest complex from β-cyclodextrin dimer and ferrocene dimer, were prepared. The particle size was 150–200 nm, and the morphology was observed using atomic force microscopy [27]. Furthermore, PEG-modified pillar [5] arene and porphyrin with pyridinium moieties were combined [28].

#### 2.1.3. ROS-Responsive

Cancer cells are subjected to more and more oxidative stress, resulting in changes in metabolic activities and carcinogenic transformation [30,31]. In particular, the level of reactive oxygen species in tumor tissue is mainly higher, which is due to the accumulation of active molecules under hypoxia in tumor tissue [32].

Recent studies have shown that special enzymes that produce reactive oxygen species have very beneficial functions in the treatment of cancer [33]. Through the self-assembly of thioketal units, photosensitizers, and Chlorin e6, a kind of ROS-responsive nanoparticle was formed and showed excellent tumor penetration and satisfied therapeutic efficacy [34]. One previous study developed integrated nanoparticles composed of poly (ethylene glycol) and polymerized methacrylate monomer loaded with β-lapachone, which is responsive to tumor ROS [35]. A kind of heparanase modified with β-cyclodextrin (β-CD) grafted heparin co-loading with doxorubicin (DOX), ferrocene (Fc), and TGF-β receptor inhibitor (SB431542) was established and successfully inhibited breast cancer metastasis, under which ferroptosis induced by ROS was essential [36].

A nano-drug composed of arginine-glycine-aspartate (RGD) conjugated with cytotoxin epothilone Bis was sensitive to ROS, showing excellent tumor selectivity and anti-cancer effect in vitro/in vivo [37]. A new ROS-responsive micelle composed of poly 10-hydroxycamptothecin and PEG, which loaded dexamethasone, was constructed, revealing an ideal anti-tumor effect [38]. Recently, phospholipid-coated Na_2_S_2_O_8_ nanoparticles that could generate new reactive oxygen species for in-situ generation of Na^+^ and S_2_O_8_^2−^ and then transform into toxic sulfate radical and hydroxyl radical (•SO_4_^−^ and •OH) were prepared. As illustrated by the scanning electron microscopy (SEM) images and TEM images, Na_2_S_2_O_8_ nanoparticles have a uniform spherical nanoflowers structure with a diameter of around 270 nm. In addition, all these effects will lead to the death of highly immunogenic cells and regulate the immunosuppressive TME, inhibiting tumor metastasis and recurrence (Figure 4) [39]. Nanoscale metal-organic frameworks (nMOFs) have made much progress in radiotherapy, photodynamic therapy, and chemodynamic therapy via nMOFs-mediated ROS generation [40]. In a previous study, Fe-metal organic framework nanoparticles were formed to generate ROS and induced cancer cell death by diminishing endogenous substances in the TME; the killing rate of tumors is up to 96.65% [41]. As for functional inorganic nanoparticles, they are also widely used because of their wide variety, precise size control, and stable function. The most widely used inorganic nanoparticles include mesoporous silica nanoparticles (MSNs) and noble metals (typically Au). For example, in a previous study, doxorubicin was loaded into the MSN pore capped with cerium oxide nanoparticles (COPs), the pre-oxidant property of COPs raised ROS levels, and a synergistic effect of drug and nanoparticle was achieved in the cancer treatment [42].

#### 2.1.4. Hypoxia-Responsive

It is well-known that an incomplete vascular network and limited oxygen diffusion distance exist in solid tumors (200 μm). Hypoxia is a unique pathological feature for 50–60% of solid tumors [43,44,45].

Self-assembled hypoxia-reactive carboxymethylglucan nanoparticles (CMD NPs) were established to promote the selective release of hydrophobic drugs in tumors [46]. Manganese dioxide nanoparticles (MnO_2_ NPs) toward hydrogen peroxide (H_2_O_2_) for the simultaneous production of O_2_ and regulation of pH were proposed to prevent tumor hypoxia and inhibit tumor growth and proliferation of breast tumors [47].

One previous study reported a platform composed of hyaluronic acid (HA)-stabilized CuMnOx nanoparticles (CMOH) and indocyanine green (ICG) for hypoxic tumor therapy and their thermally amplified catalytic activity and TME regulation ability [48]. One study reported a kind of PEG-camptothecin(CPT)-2-(piperidin-1-yl)ethyl methacrylate nanoparticle, which was hypoxia-responsive and showed tumor-suppressed effect [49]. Moreover, one study reported a hypoxia-responsive nanovesicle could enhance the efficacy of sonodynamic therapy (SDT) by generating sufficient ROS in tumors. Furthermore, SEM/TEM images showed that the morphology of the nano-assemblies had a uniform vesicular structure with an average diameter of around 129 ± 16.3 nm [50]. A new type of polymer micelles to sense hypoxia in tumors was constructed, in which the drug was released and caused immunogenic cell death through chemotherapy and photothermal therapy, which was effective in the treatment of advanced breast cancer. Characterized by TEM, the polymer micelles were spherical and monodisperse with a diameter of 84 nm (Figure 5) [51].

#### 2.1.5. Enzyme-Responsive

Tumor tissue has diverse enzyme expression profiles, which is helpful in developing efficient enzyme-responsive nano-drug delivery systems and can realize the fascinating physicochemical properties of different materials on the nanoscale [52,53].

Recently, a microneedle comprised of an anti-programmed death-ligand 1 antibody (aPD-L1) and cold atmospheric plasma therapy was also reported to induce immunogenic tumor cell death [54]. Fe^3+^ ion and naturally derived tannic acid could form sorafenib NP, which could inhibit the GPX4 enzyme for ferroptosis initiation and eventually tumor elimination [55]. One study reported that legumain-responsive gold nanoparticles (AuNPs) could lead to enhanced accumulation of doxorubicin (DOX) and hydroxychloroquine at the glioma site of patients, possessing therapeutic efficacy [56]. As reported in a previous study, carboxylesterase-responsive folate-decorated albumins into a nanocluster (FHP) confirms that it was reported to be enzyme-triggered and effective for precise cancer theranostics [57]. Matrix Metallopeptidase 2(MMP-2) responsive and RGD-peptide-modified liposome consisting of pirfenidone (an anti-fibrotic agent) and gemcitabine (a chemotherapeutic drug) was effective in pancreatic stellate cells model in BALB/c nude mice [58]. In another study, treatment with a therapeutic drug carrier loaded with monomethyl auristatin E (MMAE) via intravenous injection, MMAE targeted albumin where extracellular β-glucuronidase was overexpressed, achieved outstanding therapeutic efficacy on pancreatic tumor mice (Figure 6) [59].

#### 2.1.6. Temperature-Sensitive

The temperature of tumor tissue is usually 1–2 °C higher than normal cells, which is called hyperthermia [60,61]. Temperature is beneficial as an external stimulus in nanoparticle design. Some advantages include low toxicity and better control of cancer drug dose and localization [62].

Some types of temperature-response carriers include liposomes, polymer micelles, dendrimers, etc. [63]. Thermoresponsive micelles based on PEG-[poly(caprolactone), PCL]-PEG loaded with phenylalanine ammonia lyase showed an excellent anti-tumor effect in colorectal cancer. The molecular weight and polydispersity of the polymer were 5392 and 1.345, respectively [64]. In another study, three-arm star-shaped s-P random copolymers were modified with folic acid to make them active and targeted. The micelles were all spherical in shape, which was observed by TEM and the average distribution size was between 35 and 110 nm [65]. It was reported that using this polymer micelle system, loaded DOX will be released when the temperature reaches lower critical solution temperature (LCST) (39.2 °C), and the polymer shell will quickly release the drug at only a slightly higher temperature (40 °C) (Figure 7) [66].

#### 2.1.7. Ultrasound-Sensitive

As an external stimulus, ultrasound is popular in nano-drug research because of its non-invasive, non-ionizing radiation as well as easy adjustment of its tissue penetration depth and frequency [67]. Ultrasound can release drugs from responsive nanocarriers [68], which could increase the permeability of biological barriers by increasing the temperature to increase the absorption, release, and production of cavitation bubbles [69].

The liposome is a mature multifunctional drug delivery system. For example, under the stimulation of hyperthermia caused by ultrasound, a temperature-sensitive liposome achieved complete regression of breast cancer in mice [70], which was also applied in a human breast cancer xenograft mouse model and achieved a good effect [71,72,73]. As reported in one previous study, focused ultrasound sonication with microbubbles (MBs) could improve delivery efficiency and significantly enhance DOX accumulation [74]. What is more, low-dose focused ultrasound hyperthermia significantly enhanced the pegylated liposomal doxorubicin delivery into brain tumors and showed a promising anti-tumor effect [75]. Liu et al. showed that a nanoreactor was designed by immobilizing catalase in the large opening of mesoporous organosilicon nanoparticles (MOS). The immobilized enzyme catalyzes the decomposition of H_2_O_2_ into O_2_ molecules in a controlled manner, even when compared with 10 μm bubbles can also be produced continuously when incubated with H_2_O_2_. The bubbles generated in situ significantly enhanced the echo contrast and acted as cavitation nuclei, reducing the ultrasonic dose required for evident coagulative necrosis to 80 W. TEM images revealed a spherical shape with a diameter of 140 nm. This structure can also be used as a probe for ultrasonic diagnosis of tissue oxidative stress (Figure 8) [76].

#### 2.1.8. Magnetic Field-Sensitive

As a safe type of stimulus, magnetic fields are prevalent in nano-drug delivery in the therapy of tumors [78]. Besides treatment, magnetic field-sensitive nanoparticles were also used in diagnostic imaging [79]. A recent study used targeted magnetic carriers to deliver the DOX into the bladder wall, which caused more significant accumulation and provided site-specific delivery of drugs [80]. One recent study revealed that biomimetic magnetic Fe_3_O_4_-SAS@PLT nanoparticles were effective in curing non-inflamed tumors for ferroptosis with immunotherapy [81]. Magnetic field-induced hyperthermia could be used in the nano-drug treatment of bladder cancer in a hyperthermia-responsive manner [82,83,84]. Oxygen (O_2_) and nitric oxide (NO) were co-delivered through ultrasound-responsive nanoparticles to enhance SDT and immune response, revealing an excellent immunosuppression reversion and activation of immune response for cancer immunotherapy (Figure 9) [85].

#### 2.1.9. Light-Sensitive

Light-responsive nanoparticles are popular for their convenient use at different wavelengths, such as ultraviolet (UV) [86], visible [87], and near-infrared (NIR) [88]. Nowadays, UV is widely used in functional nanoparticle design. Compared with UV, the deeper penetration of NIR is effective for deep tissue therapy [89].

Light-controlled drug delivery was first reported in 2010 [90]. The UV light irradiation could induce the burst release of hydrophobic NR molecules from the nanoparticle [91]. In another study, a copolymer system consisting of poly(ethylene oxide) (PEO) and poly(2-nitrobenzyl methacrylate) (PNBM) was sensitive to UV-light and led to the light-controlled release of payload drugs [90]. Compared with UV, infrared ranges from 600–900 nm, has deeper penetration, and is more effective in treating cancer in the deep part of the tissue. NIR-responsive systems are another main light-responsive system [92]. As an FDA-approved fluorescent dye, ICG is widely used in cancer therapy, which could absorb NIR light into heat [93]. Additionally, doxorubicin and ICG loaded in PLGA-based and dual-modality imaging guided chemo-photothermal nanoparticles showed faster release under NIR irradiation; this nanoparticle had a diameter of around 200 nm with good fluorescence stability. Doxorubicin release was stimulated by heat, and nanoparticle penetration of the tumor was improved after NIR irradiation, this research provided a promising strategy for early diagnosis and therapy for cancer [94]. In a previous study, the dual-responsive supramolecular prodrug complexes (SPCs)-based self-assemblies were achieved by utilizing UV and pH stimuli, with easier internalization into the cancer cells and therapy efficacy (Figure 10) [95].

### 2.2. Surface Conjugation with Targeting Ligands

The active target of cancer cells based on monoclonal antibodies, peptides, and aptamer has aroused significant interest in the field of targeting ligands [4].

#### 2.2.1. Monoclonal Antibodies

Antibody-drug conjugates are monoclonal antibodies conjugated to cytotoxic agents [96]. Monoclonal antibodies (MAbs) are macromolecules widely used as targeting ligands to various types of nanoparticles, such as SPIONs [97], QDs [97], liposomes [98], and Au nanocages [99]. However, their bulky size and constant redundant region may limit their use. As reported in a previous study, nanoparticles formed by 2-methoxy-estradiol (2-ME) based on anti-human epidermal growth factor receptor 2 (HER2) antibody-modified BSA were validated in targeted cancer therapy; this system was prepared using the desolution method and was proven effective in retaining the immunospecificity of the anti-HER2 antibody for the targeted cancer therapy [100]. Furthermore, in a previous study, PTX was absorbed on graphene oxide nanosheets and then conjugated with vascular endothelial growth factor (VEGF) to form the targeted nanoparticles, which showed remarkable potential in photothermal controllable tumor treatment [101].

#### 2.2.2. Peptides

Due to the bulky size of MAbs, peptides represent a viable targeting moiety with relative flexibility and overcome the disadvantages of using MAbs [102]. Peptide−drug conjugates (PDCs) are increasingly recognized in targeted drug delivery [103]. For example, RGD peptides were intimately connected with integrin [104]. Nanoparticles composed of RGD conjugation with superparamagnetic iron oxide nanoparticles (SPIONs) possessed better targeting affinity and specificity [105]. In one previous study, a biocompatible conjugate consisting of a fatty acid-substituted dextran decorated with cyclo[RGDfK(C-6-aminocaproic acid)] cRGDfK peptide could be used as a candidate for conventional PEG [106]. Recently, gemcitabine (GEM) and amphiphilic peptide conjugation were effective in breast cancer therapy, this system was formed via self-assembled behaviors, and the stability of GEM could be maintained during the circulation and accumulated in the tumor site, which broadened the application of GEM for breast cancer therapy [107]. Tumor vascular endothelial cells (tVECs) were targeted with cRGD-functionalized polyplex micelle loading anti-angiogenic protein encoding pDNA for anti-tumor activity in human pancreatic adenocarcinoma tumor-bearing mice by noting that tVECs abundantly express αvβ3 and αvβ3 integrin receptors [108,109]. Moreover, the application of cell-penetrating peptides (CPPs) based on nanoplatforms in cancer treatment has also been worth attention since their identification 25 years ago [110]. Nanoparticles could be functionally attached to CCPs to achieve good therapeutic efficacy. For example, a nanoparticle against CapG to a series of CPPs showed a good potential for metastasis in breast cancer [111].

#### 2.2.3. Aptamers

Aptamers are complex three-dimensional structures composed of short, single-stranded, synthetic nucleic acid oligomers, DNA or RNA, with some good characteristics for their high affinity and specificity, easy synthesis, low molecular weight, and lack immunogenicity [112]. As ligands, conjugation of aptamers to nanoparticles has been reported in a series of previous studies [113]. The combined use of cisplatin in liposomes conjugated with aptamers has been reported where the nucleolins (NCL) were the target of aptamers. Aptamer-conjugated liposomes were formed by cholesterol incorporation and hydration, which showed strong anti-proliferative activity in breast cancer cells overexpressing NCL [114].

### 2.3. Large Molecules-Based Therapy

#### 2.3.1. Nucleic Acid-Based Therapy

Nucleic acid-based therapy is a technology that transfers nucleic acids, including plasmid DNA, mini vector DNA, siRNA, etc., to the nucleus of diseased cells or tissues for the gene therapy of cancers [115,116,117]. Furthermore, gene therapy focuses on the mutated genome of the tumor cells [118], aiming to restore instead of kill cells [119].

Nowadays, there are two methods, viral and non-viral, in gene therapy. Safety concerns have limited the routine use of viral vectors. In contrast, a non-viral gene delivery method is preferred.

Poly(*N*-isopropylacrylamide) (PNIPAM) is the most extensively used in gene delivery systems [120]. The polyplex micelles consisting of PNIPAAm and therapeutic plasmid DNA (pDNA) could prolong blood circulation and suppress tumor growth in H22 tumor-bearing mice [121]. Nanoparticle-based delivery of small interfering RNAs (siRNAs) has also been reported to possess an anti-proliferative effect [122]. A biodegradable and redox-sensitive nanocarrier consisting of solid poly (disulfide amide) (PDSA)/cationic lipid core and a lipid-PEG shell for siRNA delivery was proven to have a good therapeutic effect [123]. Liposomes targeting the interleukin 12 (IL-12) gene in a non-viral manner could induce an immune response and achieve good therapeutic efficacy [124].

Additionally, thermosensitive nanocarriers have also been used in gene transfection. For example, PEG polymers with grafted PEI chains were used to improve transfection efficiency [125]. In another study, the core GCP was replaced with transactivator of transcription (TAT)-modified AuNPs loaded with Cas9/sgRNA-Plk1 plasmid to obtain multifunctional clustered, regularly interspaced, short palindromic repeats-associated protein/single guide RNA-polo-like kinase 1 (Cas9/sgRNA-Plk1) plasmid-loaded multifunctional nanocarriers [126]. Under the stimulation of NIR, CRISPR/Cas9 plasmid delivery has been successful in genome editing (~20%) of MTH1 in HCT 116-GFP tumor models (Figure 11) [127]. At present, lipid nanoparticle-mediated mRNA delivery for chimeric antigen receptor (CAR) T cell therapy is promising. However, the viral delivery vectors caused severe side effects. Ionizable lipid nanoparticles (LNPs) could achieve safe and stable delivery of mRNA to human T cells for the activation of functional proteins to enhance the efficacy of CRA T treatment [128].

#### 2.3.2. Protein-Based Drug Delivery

Natural biological molecules usually form protein-based nanoparticles with biocompatible, biodegradable, and non-antigenic properties and can be functionalized with cell-targeting groups or ligands [129,130,131]. Gelatin is a protein obtained from the hydrolysis of collagen [132]. One previous study in dogs with bladder cancer revealed that gelatin nanoparticles loaded with paclitaxel (PTX) had a good therapeutic efficacy [133]. Drug molecules conjugated with proteins were used for cancer therapy [134]. Hollow mesoporous silica capsules have garnered significant attention as protein delivery vehicles [135]. For example, Fluorescein isothiocyanate (FITC)-labeled proteins were loaded into the nanoparticles to achieve efficient therapeutic efficacy [136], evidenced by several examples of protein delivery in liposomes [137]. Moreover, magnetic field-responsive protein conjugation nanoparticles were also reported to be an efficient system for brain tumors [138].

### 2.4. Others—Hydrogel

Hydrogel is a polymer with a three-dimensional, hydrophilic, and cross-linking network, capable of retaining a large amount of water or physiological fluids [139]. Injectable biodegradable hydrogels have broadened novel methods of cancer treatment [140,141].

In cancer therapy, hydrogels provide a platform for drug combinations. For example, an injectable DNA hydrogel assembled by chemo drug-grafted DNA in a previous study showed excellent anti-tumor efficacy and represented a promising adjuvant therapy in cancer treatment [142]. Additionally, thermosensitive PPZ hydrogel loaded with PEGylated cobalt ferrite nanoparticles showed a fantastic therapeutic efficacy in the breast cancer mice model [143]. Furthermore, a unique “Jekyll and Hyde” nanoparticle–hydrogel (NP-gel) hybrid platform was designed to load DOX, leading to a good anti-recurrence efficiency and low toxicity [144]. Based on the hyperthermia caused by ultrasound, a magnetic hyperthermia responsive hydrogel comprised of silk fibroin and iron oxide nanocubes was effective in the 4T1tumor-bearing mice model [145]. A light-responsive hydrogel consisting of Ag_2_S and Fe-doped bioactive glass was proven to inhibit tumor growth (Figure 12) [146]. Moreover, a hydrogel composed of indoleamine 2,3-dioxygenase-1, and chemotactic CXC chemokine ligand 10 was designed and delivered in the regressing postresection tumor relapse, especially for its immunity modulation in glioblastoma multiforme [147]. Moreover, PEG hydrogel was constructed as a co-delivery system, where the PEG chain and cyclodextrin determine the transition temperature [148].

## 3. Preclinical to Clinical Transformation

As for the development of functional nanoparticles in cancer therapy, although great progress has been made which was summarized in Table 1, there are still several challenges in safety and drug release which need to be overcome if they are to be transformed into clinical use [149]. Understanding the in vitro and in vivo delivery is a significant step for the application of functional nanoparticles in clinics. The nanoparticles must be firstly evaluated at the cellular level and, further, at tissue, organ, and body levels. During the intracellular transport process, the main steps, including endocytosis, intracellular transport, escape, and degradation, need to be considered. While nanoparticle–cell interaction is the main concern in the in vitro application, the most important thing for in vivo application is achieving the targeted delivery to the tumor site. The main processes, including the EPR effect, passive tumor targeting, active targeting, pharmacokinetics, biodistribution, and clearance, are vital [150,151,152,153,154]. Learning from multidisciplinary knowledge, such as computational modeling, is significant in this progress [155,156]. Computational modeling could be used in the prediction and monitoring of releasing and interaction behaviors of nanoparticles in the body despite the limitation of translating results from computational models into clinical practices on account of the complexity and heterogeneity of clinical tumors [157]. Furthermore, drug release is not controllable to some degree due to the heterogeneity from patient to patient. More advanced approaches need to be proposed to be used in the nanoparticle-controlled release. For example, the microchip is a relatively competent device in this field [158].

## 4. Discussion

Functional nanoparticles are promising for improving cancer treatment significantly. Functional nano-drugs are designed as a platform in response to TME-related stimuli, such as acidic pH and increased secretion of GSH [159,160], or external temperature, ultrasound, magnetic field, and light [161]. In addition, surface conjugation and large molecule delivery are also applicable in functional drug delivery. Furthermore, nano-drugs have versatility, combining diagnosis and treatment characteristics. In recent years, functional nanoparticles have become increasingly important in cancer immunotherapy because of the delivery of tumor-related immunomodulators [162,163]. This research topic on functional nanoparticles in cancer therapeutic therapy is exciting because of the current progress; however, there are still many problems to be solved on the road to the wide application of functional nanoparticles.

As for challenges, the biggest problems of these functional nano-drugs are their instability during blood circulation, toxicity, low renal clearance, shallow penetration depth, uncontrollable emission under stimulation signals, low uptake, and accumulation in the cancer cells/tissues [164]. Thus, it is vital to understand the interaction between functional nanoparticles and biological systems at cell, tissue, organ, and body levels. For decades, people have been attempting to improve the therapeutic effect, but reducing toxicity in the development of cancer nano-drugs has not explicitly been solved [165,166]. When functional nanoparticles could modulate the toxicity of anti-cancer drugs, a new direction, and better opportunities were provided for the translation. Multidrug resistance (MDR) is a sophisticated process involving multiple mechanisms, such as efflux pump-mediated/dependent MDR, tumor cell heterogeneity, clonal selection, and expansion. The present functional nanoparticles could be designed based on the tumor genetic profiles via a series of nanoplatforms, including liposomes, dendrimers, and metallic nanoparticles, which could be promising for overcoming cancer drug resistance and achieving a better effect [167,168,169]. The use of redox enzymes still needs a lot of research [170]. For example, light-responsive nanoparticles were limited by penetration depth. A fundamental obstacle lies in developing optimal methods for loading and releasing nanoparticle drugs. The encapsulation of drug molecules must be stable during the idle period. Still, once it enters the tumor site, it will be activated to release the drug [171,172]. Developing advanced methods for overcoming the above problems to improve the effectiveness of cancer treatment is a principal goal.

In recent years, functional nanoparticles have also been applied in cancer immunotherapy because of their delivery of tumor-associated immunomodulatory agents/antigens to activate dendritic cells for the elimination of cancer cells through this immunomodulatory process. The induction of cellular immune responses at mucosa is now of great interest in vaccine development targeting mucosal pathogens. For example, an effector memory T cell-based mucosal nanoparticle vaccination could promote robust T cell responses for protection [173]. Furthermore, combined with other agents, immune checkpoint inhibitors targeted to immune inhibitory receptors such as CTLA-4, PD-1, and PD-L1 could achieve a more durable efficacy and excellent safety [174,175,176].

In addition, it is essential to develop an in vitro/in vivo test platform for the practical evaluation of functional nanoparticles. Notably, there are significant differences between in vitro and in vivo tumor models, as well as between animals and humans. To date, most nanoparticles work well in vitro but fail to test in a more complex in vivo microenvironment. A few years ago, the exciting concept of “organ-on-chip” was proposed [158]. In this method, 3D miniaturized in vitro human tissues/organs were created through perfusion culture on a microfluidic chip and connected to form a multi-organ human simulation platform, which could be used to test drugs and nanoparticles and predict their behavior in vivo.

As for in vivo drug delivery, several issues need to be considered, including the EPR effect, active targeting, clearance by the MPS, renal clearance and pharmacokinetics and biodistribution, biocompatibility, and biodegradation, which were shown in detail in another review [177]. The clearance rate of nanoparticles is a critical issue in the design of nanoparticles [178]. Although most nanoparticles can degrade after they are delivered as drugs, many other systems cannot degrade quickly, but their potency is too strong to be ignored. In these cases, it is necessary to design according to its physiological correlation with cancer and normal tissues [179].

Due to the high heterogeneity and complexity of the TME, nanoparticles’ targeting and therapeutic ability are often quite limited, including a variety of subsets [180]. Moreover, there are similar characteristics between tumors and normal tissues [181]. Two schemes have been designed to solve this problem, involving a variety of delivery/target mechanisms. Co-delivery of multi-drugs via nanoparticles can be designed to target multiple anti-tumor parts simultaneously [182]. Similarly, better treatment efficacy can be achieved via antibody conjugation against one tumor type for tumor recognition [183].

In recent years, carrier-free nano-drugs have contributed to the progress of various treatment methods [184,185]. The advantages of anti-cancer medications combined with other chemotherapeutic drugs, photosensitizers, photothermal therapy, immunotherapy, or genetic drugs have been proven. Finally, this paper introduces the prospects to emphasize the challenges and possible solutions of the currently developed carrier-free nano-drugs in clinical application, which may have significance for designing effective carrier-free solutions in the future.

Apart from the treatment characteristic, the theragnostic agents used for cancer diagnosis and personalized treatment is also important. The concept of theranostics emerged around 2002 to combine diagnostic assays with therapy. Molecular imaging was applied in the cancer treatment for its tumor target specificity and minor damage to the normal tissue. There is a growing number of nanoscale probes that have been developed for imaging modality, therapeutic cargo, and the target [186]. For example, the prostate-specific membrane antigen-targeted nanoplexes carrying imaging reporters, siRNA, cDNA, and prodrug enzymes in the cancer diagnosis have been reported [187,188]. As a young field for theranostics, some issues, including side effects of probes, synthesis, and translation, still need to be solved. Moreover, functional nanoparticle strategies contributed a lot to this cancer diagnosis and treatment [186]. Importantly, theranostics could reveal the actual characteristics of cancer and the side effects in patients; therefore, this feedback could help for personalized treatment.

In summary, with the technical revolution of nanotechnology, more stable and biocompatible functional nano-drug delivery systems will be developed and better used in cancer diagnosis and treatment.

## 5. Future Directions

Facing the main challenges with nanoparticles in cancer treatment, some aspects still need attention. First, reducing side effects and improving the delivery efficiency of nanoparticles through interdisciplinary approaches, such as computational modeling. Secondly, more attention needs to be paid to controlling the loading and releasing of nano cargos. Thirdly, a multifaceted evaluation platform is worth efforts to be built up for a comprehensive study and monitoring for in vitro/in vivo interaction between nanoparticles and cells/whole organism.

## Figures and Tables

**Figure 1 pharmaceutics-14-01682-f001:**
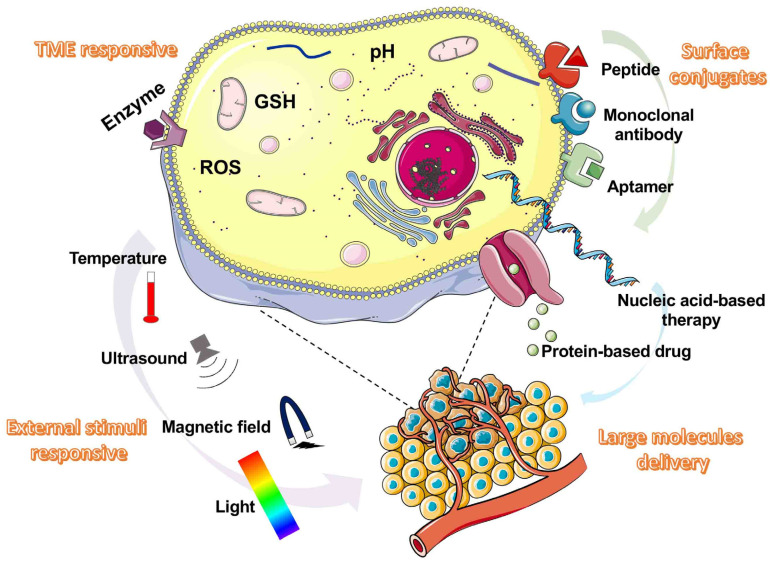
Main functional nanoparticles types in cancer therapy.

**Figure 2 pharmaceutics-14-01682-f002:**
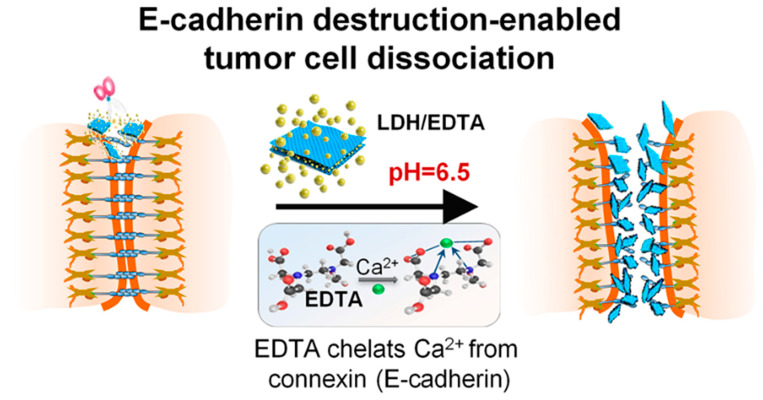
A scheme of tumor cell dissociation induced by E-cadherin pH-responsive nanoparticle in low colorectal tumor. Reprinted with permission from Ref. [20]. Copyright 2022, Shi, J.

**Figure 3 pharmaceutics-14-01682-f003:**
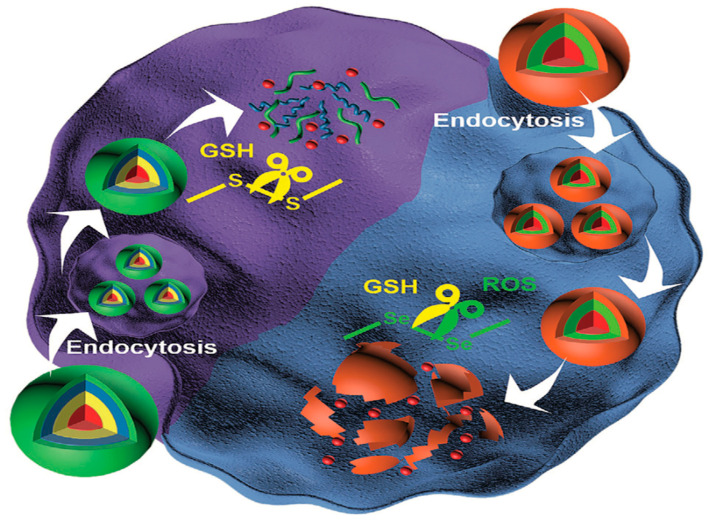
The scheme for GSH-responsive nanoparticle delivery system. Reprinted with permission from Ref. [29]. Copyright 2020, Wu, J.

**Figure 4 pharmaceutics-14-01682-f004:**
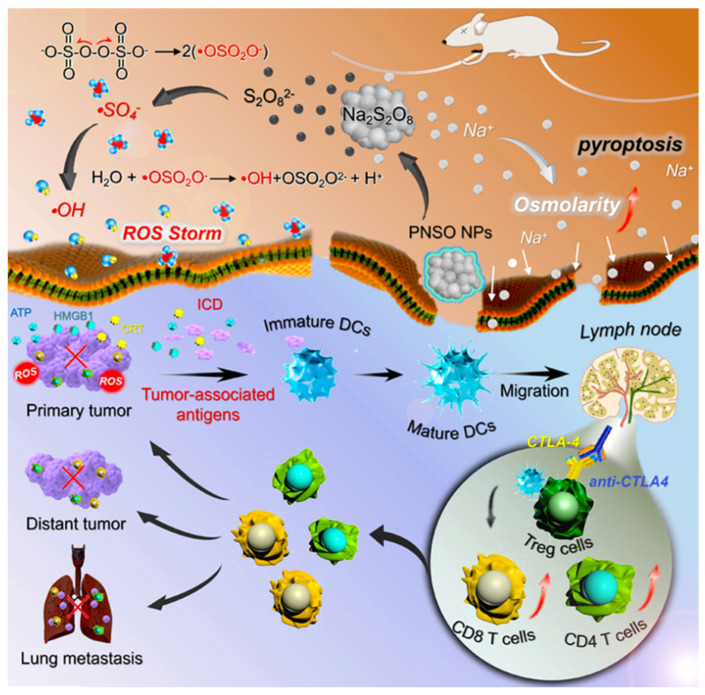
Scheme of therapeutic efficacy and mechanism of ROS-responsive Na_2_S_2_O_8_ nanoparticles. Reprinted with permission from Ref. [39]. Copyright 2020, Zhang, H.

**Figure 5 pharmaceutics-14-01682-f005:**
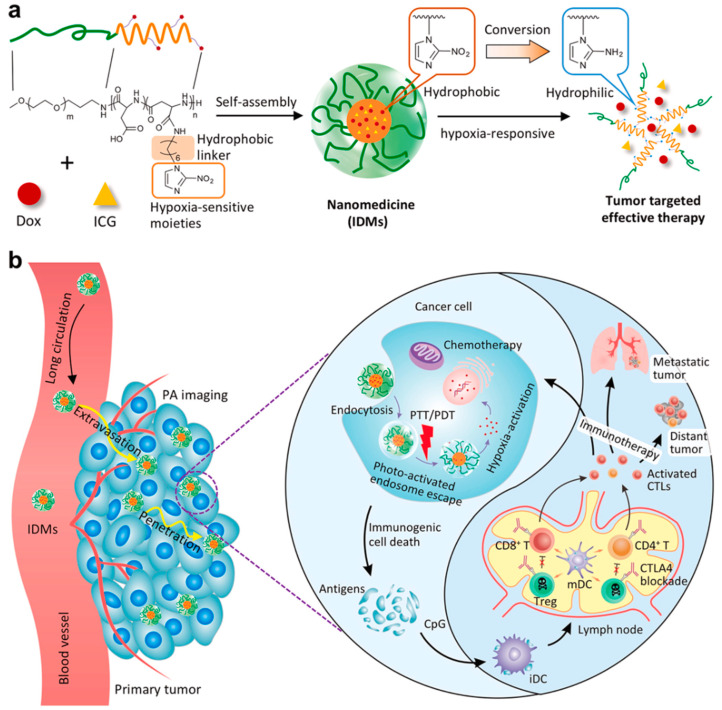
Scheme of hypoxia-responsive micelles in advanced breast cancer. (**a**) Nanoparticle formulation; (**b**) The mechanism of the hypoxia-responsive nanoparticle in treating advanced breast cancer. Reprinted with permission from Ref. [51]. Copyright 2021, Mi, P.

**Figure 6 pharmaceutics-14-01682-f006:**
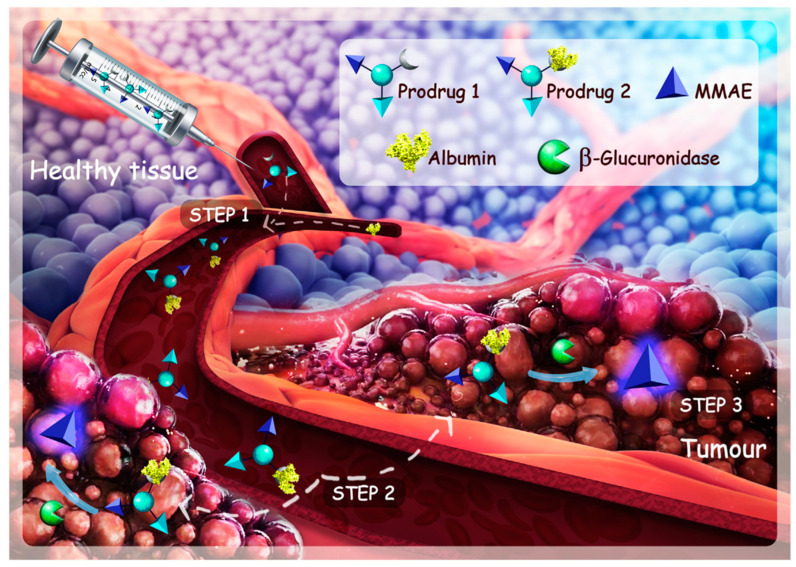
The main GSH-responsive progress in the tumor. Reprinted with permission from Ref. [59]. Copyright 2017, Papot Sebastien.

**Figure 7 pharmaceutics-14-01682-f007:**
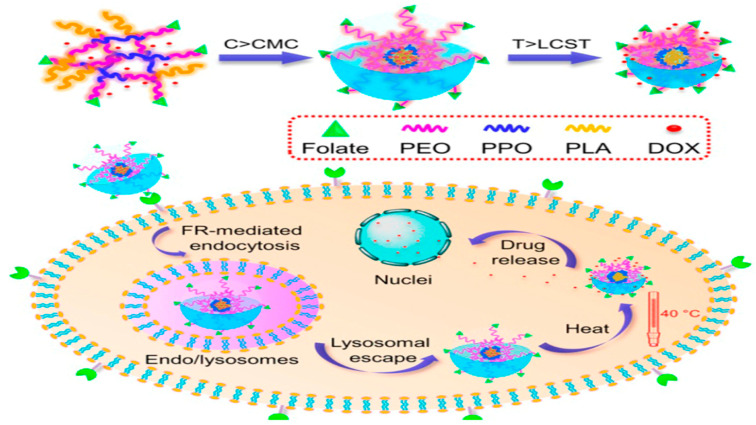
Scheme of temperature-responsive micelle formation progresses in the tumor tissue responsive to temperature. Reprinted with permission from Ref. [66]. Copyright 2014, Zhou, S.

**Figure 8 pharmaceutics-14-01682-f008:**
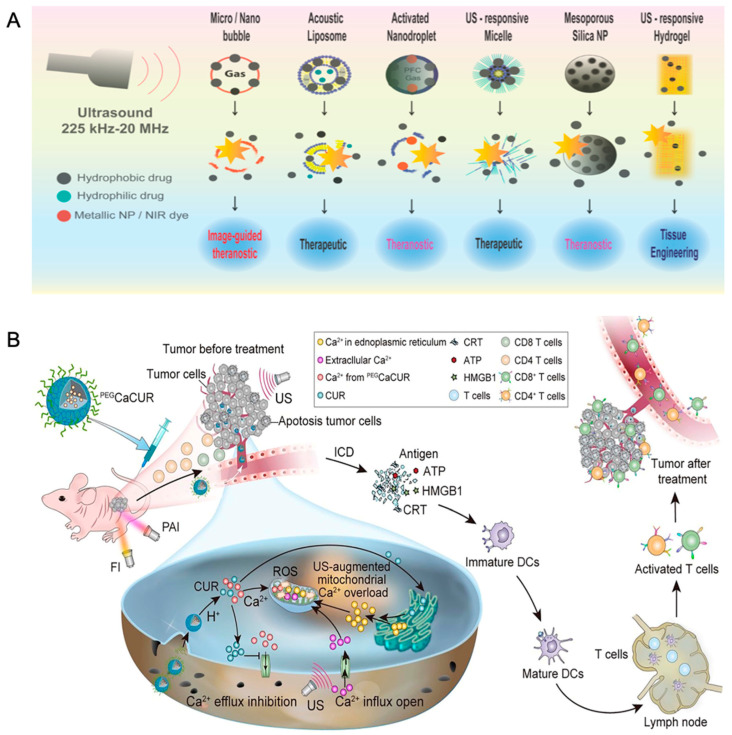
(**A**) Scheme of ultrasound-responsive nanoparticles for therapeutic applications. Reprinted with permission from Ref. [77]. Copyright 2014, Banerjee, R.; (**B**) Scheme of Photoacoustic/Fluorescence Dual-responsive nanoparticles induced mitochondria calcium ion overload leading to immunogenic cell death. Reprinted with permission from Ref. [76]. Copyright 2016, Chen, X.

**Figure 9 pharmaceutics-14-01682-f009:**
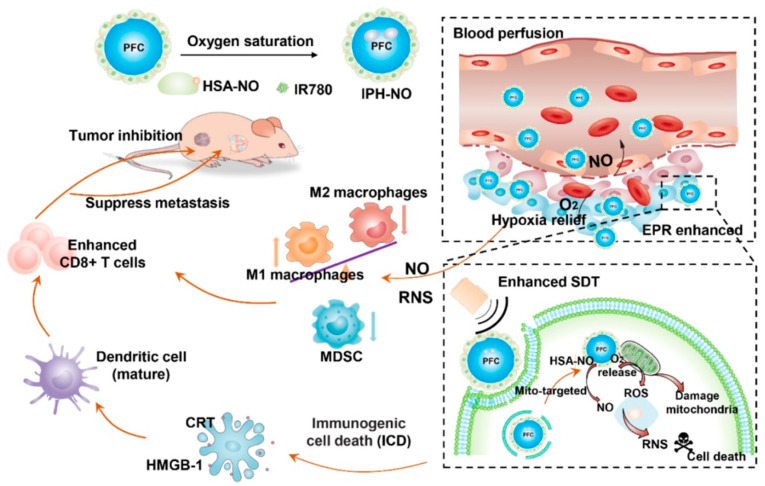
Scheme of magnetic responsive nanoparticles for cancer therapy, where immune activation, inhibition of primary tumor, and metastasis to the lung were the main underlying modulating progress. Reprinted with permission from Ref. [85]. Copyright 2021, Guo, H.

**Figure 10 pharmaceutics-14-01682-f010:**
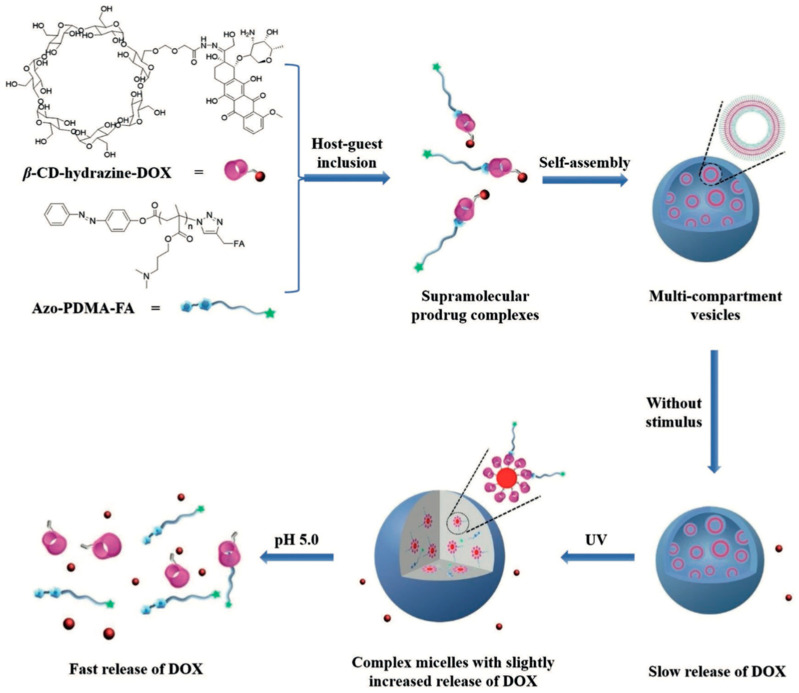
The drug release progress under UV stimulus. Reprinted with permission from Ref. [95]. Copyright 2018, Tian, W.

**Figure 11 pharmaceutics-14-01682-f011:**
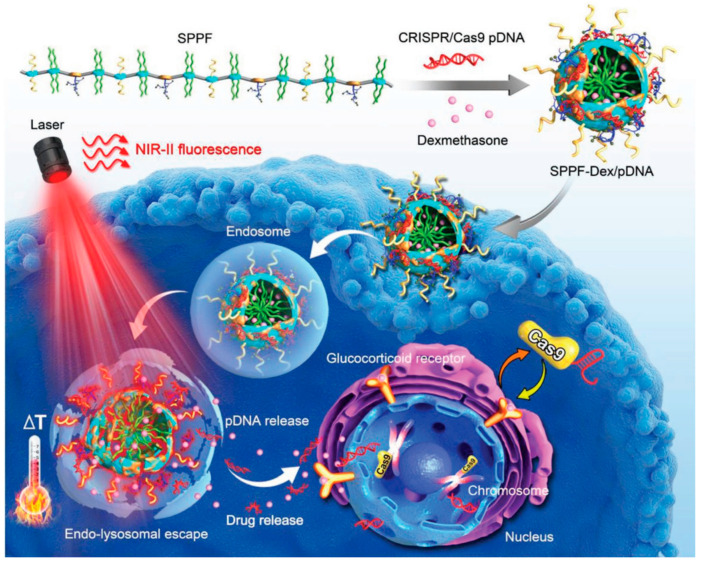
Gene delivery based on CRISPR/Cas systems. The therapeutic efficacy and mechanism of the nanoparticle in HCT 116-GFP tumor models. Reprinted with permission from Ref. [127]. Copyright 2019, Chen, X.

**Figure 12 pharmaceutics-14-01682-f012:**
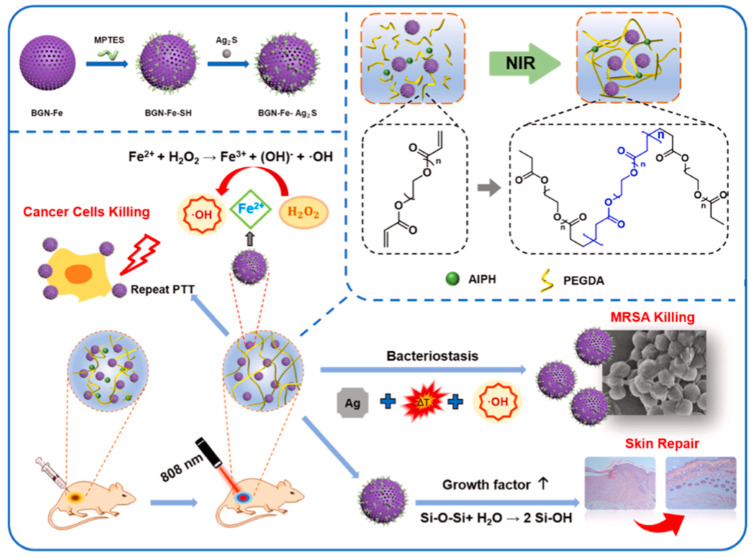
A scheme of BGN-Fe-Ag2S and PBFA hydrogel preparation and its anti-tumor and anti-infection effect. Reprinted with permission from Ref. [146], Copyright 2022, Zhao, Y., especially the modulating effect of the hydrogel on tumor immunity and recurrence.

**Table 1 pharmaceutics-14-01682-t001:** Functional nanoparticles in clinical translation.

Formulation	Active Ingredient	Trademark	Label Indication	Status
Albumin NP	Paclitaxel	Abraxane	Breast, lung, and pancreatic cancer	Approved
Inorganic nanoparticles	Ferric oxide	Feraheme	Iron deficiency anemia (chronic kidney disease)	Approved
	Silicon dioxide	Cornell Dots	Imaging: melanoma, brain tumor	Preclinical trial
	Silica dioxide-gold	AuroLase	Lung cancer (photothermal therapy)	Preclinical trial
Liposomes	Doxorubicin	Doxil	Kaposi sarcoma, breast cancer, ovarian cancer, multiple myeloma	Approved
	Daunorubicin	DaunoXome	Kaposi sarcoma	Approved
	Doxorubicin	Myocet	Metastatic breast cancer	Approved
	Paclitaxel	Lipusu	Ovarian cancer, metastatic gastric cancer	Approved
	Mifamurtide	MEPACT	Osteosarcoma	Approved
	Vincristine sulfate	Marqibo	Acute lymphoblastic leukemia	Approved
	Irinotecan	Onivyde	Metastatic pancreatic cancer	Approved
	Cytarabine/daunorubicin (5:1)	VYXEOS	Acute myeloid leukemia	Approved
	Cisplatin	Lipolatin	Non-small cell lung cancer	Clinical trial phase 3
	Irinotecan	IHL-305	Advanced stage of solid tumor	Clinical trial phase 1
Nanoparticles	Paclitaxel	DHP107	Advanced gastric cancer	Approved
	Doxorubicin	Transdrug^®^	Hepatocellular carcinoma	Approved
	Docetaxel	BIND-014	Advanced non-small cell lung cancer	Clinical trial phase 2
	Camptothecin	CRLX101	Advanced non-small cell lung cancer	Clinical trial phase 2
	Anti-RRM2 siRNA	CALAA-01	Solid tumor	Clinical trial phase 1
	Paclitaxel	Nanoxel^®^	Advanced stage of breast cancer	Clinical trial phase 1
	Paclitaxel	PICN	Metastatic breast cancer	Approved
Polymer-drug conjugate	Asparaginase	Oncaspar^®^	Leukemia	Clinical trial phase 3
	Paclitaxel	Xyotax^®^	Breast cancer, ovarian cancer	Clinical trial phase 3
	Paclitaxel	Taxoprexin^®^	Solid tumor	Clinical trial phase 2
	Camptothecin	XMT-1001	Non-small cell lung cancer	Clinical trial phase 1
	Paclitaxel	Genexol-PM	Breast cancer, lung cancer	Approved
Polymeric micelles	Paclitaxel	Nanoxel	Breast cancer, ovarian cancer	Approved
	Paclitaxel	Paclical	Ovarian cancer	Approved
	Doxorubicin	Themodox^®^	Breast, lung cancer	Clinical trial phase 3
	Paclitaxel	Genexol^®^-PM	Breast cancer, lung cancer, pancreatic cancer	Approved
	Paclitaxel	Paclical^®^	Ovarian cancer	Approved
	Paclitaxel	NK105	Gastric cancer	Clinical trial phase 3
	Oxaliplatin	NC-4016	Solid tumor	Clinical trial phase 1
	Cisplatin	NC-6004	Pancreatic cancer	Clinical trial phase 3
	7-ethyl-10-hydroxycamptothecin	NK012	Triple-negative breast cancer	Clinical trial phase 2

## Data Availability

Not applicable.

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
