# Peer review of "Functional Nanoparticles for Enhanced Cancer Therapy"

_pharmaceutics, 2022, doi:10.3390/pharmaceutics14081682_

Round 1

Reviewer 1 Report

This paper addresses the use of nanoparticles in cancer treatment. The authors explained the different strategies used to prepare stimuli-responsive and other varieties of nanoparticles. Overall, this paper has tried to cover different fields but still needs to be thoroughly reviewed.

With that, there are a few areas of concern:

1.      In 2.1.2. section, lines 100-102, the author explained the ? - cyclodextrin dimer, ferrocene dimer- based, and PEG-modified delivery system but I am unable to find Figures 3B and 3C.

2.      In 2.1.3. ROS-responsive section, please add about the different organic and inorganic nanoparticles used in the cancer treatment.

3.      Could the authors please explain about NIR-based chemo-therapeutic delivery system in detail?

4.      Please add the details of antibodies and peptides used for cancer treatment in section 2.2. Could the author also explain the cell-penetrating peptides?

5.      Please explain the nucleic acid based LNPs delivery system for cancer treatment and the current progress made with LNPs in the gene delivery system.

6.      Please explain the theragnostic agents used for cancer diagnosis and personalized treatment.

7.      Could the author explain the functional nanoparticles used in cancer immunotherapy in detail? What approaches could be employed to improve the effectiveness of vaccines for immunotherapy? Also, please explain the recently approved immune checkpoint blockers in cancer treatment.   

8.      It would be better to list out the different functional nanoparticles and targeting moieties that are currently using in the clinical trials and clinics.

9.      Could the author explain the challenges associated with the current drug delivery systems?

10.   Could the author also explain the problem associated with in-vitro/ in-vivo translation of functional NPs?

11.   What could be the significant role in reducing systemic toxicity and overcoming drug resistance of existing nanoplatforms? Explain in detail.

12.   Please correct the typo in the 2.2 section, line 259.

Reviewer 2 Report

We need to clarify some things before we can move forward, and then a positive action can be taken.

There isn't much in-depth analysis of each section in the review article.

Ø TEM was used to assess each nanostructure made with nanoparticles. Could the author add additional statistically significant characterisations?

Ø What can we infer about placement from nanoparticles?

Ø How to apply this material is an important issue.

Reviewer 3 Report

S. Wu and his co-researchers surveyed the recent progress of a topic entitled “Functional nanoparticles for enhanced cancer therapy”. This manuscript summarizes various strategies for encapsulating drugs and macromolecules into drug delivery systems for responding to exogenous and endogenous stimuli. In addition, this manuscript provides a detailed review of the progress of functional nanoparticles in cancer therapy.  

The authors provide an appreciable review worth publishing in the MDPI Pharmaceutics. However, this manuscript needs some major and minor changes before being accepted. I recommend the authors a detailed revision addressing the following issues carefully.

1.    Line 289 and Line 290: In the following original statement, the authors stated that siRNA is a gene, and siRNA works in the cytoplasm. siRNA is not considered a gene and is not work in the nucleus. Better change the gene therapy to nucleic therapy. Plasmid DNA, minivector DNA must be sent to the nucleus, whereas mRNA and siRNA must reach the cytoplasm to show their activity. 

The original statement by the authors: Gene therapy is a technology that transfers nucleic acids, including plasmid DNA, minivector DNA, siRNA, etc. to the nucleus of diseased cells.

2.    Define what hyd is? 

           doxorubicin-hydrazone bond-PEG-folic acid

3.    Expand the following terms after their first appearance in the manuscript. 

PTX  Paclitaxel

NGO Nano graphene oxide 

CUR  Curcumin 

EDTA  Ethylenediaminetetraacetic acid 

RGD  Arginine-Glycine-Aspartate

cRGDfK  cyclo[RGDfK(C-6-aminocaproic acid)] 

MMP-2 responsive  Matrix Metallopeptidase 2

TAT  transactivator of transcription 

PNIPAAm  Poly(N-isopropylacrylamide) (PNIPAM)

Cas9/sgRNA-Plk1  Clustered Regularly Interspaced Short Palindromic  

Repeats-associated protein/single guide RNA-polo-like kinase 1 

FITC  Fluorescein isothiocyanate

SPIONs  Superparamagnetic iron oxide nanoparticles

• SO4− and • OH  sulfate radical and hydroxyl radical

LCST  lower critical solution temperature

4.    In Line 51, TME was expanded as tumor microenvironment (TME)-responsive. However, in Line 62, again acidic tumor microenvironment was used. The authors can use acidic TME.

5.    Use proper superscripts and subscripts in the manuscript.                                   Zn2 GeO4 :Mn2+ ,Pr3+  Zn2GeO: Mn2+, Pr3+

Line 231: Oxygen (O2

Line 127: S2O82−  S2O82−

Na2S2O8  Na2S2O8

6.    Do not use the acronyms if the information is given only time in the text. 

(ZGMP) was used only one time. Hence no need to use acronyms in the brackets after Zn2 GeO:Mn2+/Pr3+. Similar to the case for the following. The following acronyms increase the word count of the manuscript but no specific scientific information if it is used only one time. 

(HA@CURNPs, 

(FA-GEM-AgNPs) 

(?-CD2) (Fc2)

(HPSE) 

(pnso NPs)

(Epo B) 

(hMVs) 

(PEMA) 

(CAP) 

(PSC) 

(PLD) 

(LACP) 

(IDO1) 

(HMSCs)

(PAL)

(NLC/H(D + F + S) NPs)  What H, D + F + S are standing for

HCQ

(CE) 

7.    Many typos were observed throughout the manuscript. 

Line 72: decomposed t the  decomposed the

Line 77: which I achieved  which achieved

Line 100: ? - cyclodextrin dimer (?-CD2) and ferrocene dimer (Fc2), was

prepared  ?-cyclodextrin dimer and ferrocene dimer were prepared.

Line 212: 10 μ bubbles 10 μm bubbles

Line 214: mH2O2.  H2O2

Line 191:  39.2 ′ C  39.2 

8.    Unnecessary gaps were observed throughout the manuscript.

PEG- camptothecin  PEG-camptothecin

9.    No gap was used between words. 

Line 309: nanocarriers(LACP)   nanocarriers (LACP)

Line 149: sonodynamic therapy(SDT)  sonodynamic therapy (SDT)

10. Although it is mentioned in the abstract that there are still some challenges and limitations needed to be considered, no challenges and limitations were described in the manuscript. 

I suggest the authors create a section with the challenges and solutions of nanoparticles for enhanced cancer therapy. The following description might help the authors to introduce the section.

One of the major critical issues in systemically injected nanoparticles is nonspecific sequestration by the liver, resulting in a substantial decrease in the delivery efficiency of nanoparticles into diseased tissues (Nat Mater. 2016;15(11):1212-1221). To address this unwanted liver-mediated capture, several novel liver blockade strategies have been proposed. In contrast to the stealth coating of nanoparticles by nonionic PEG, an in vivo transient and selective stealth coating of liver scavenger sinusoidal wall has been developed using a two-arm-PEG-Oligocatiomer. This stealth coating inhibited the sequestration of nanoparticles by the liver scavenger cells, thereby augmenting the gene delivery efficiency in the target organs (Sci Adv. 2020;6(26):eabb8133). In another reticuloendothelial system blockade strategy, the RES was forced to clear the organism’s own intact red blood cells by marking them with allogeneic anti-erythrocyte antibodies. This forced antibody-mediated erythrophagocytosis increases the circulation times of nanoparticles, thereby enhancing the anti-tumor activity of nanoparticles (Nat Biomed Eng. 2020;4(7):717-731). Dosing above 1 trillion nanoparticles in mice overwhelmed Kupffer cell uptake in the liver, thereby prolonged the blood circulation half-life of nanoparticles, and ultimately increased the delivery efficiency to the tumor site and antitumor activity(Nat Mater. 2020;19(12):1362-1371).

11. Sometimes both acronyms and their expansion were used more than two times. 

Line 144 and Line 249, indocyanine green (ICG)

Line 166 and Line 307: gold nanoparticles (AuNPs) 

Line 301 and Line 116: poly(ethylene glycol) (PEG)

12. It is recommended to add the following information under section, 2.2.2. Peptides. Tumor vascular endothelial cells (tVECs) were targeted with cRGD-functionalized polyplex micelle loading anti-angiogenic protein encoding pDNA for antitumor activity in human pancreatic adenocarcinoma tumor-bearing mice by noting that tVECs abundantly express vβ3 and vβ3 integrin receptors (Biomaterials 2014;35(20):5359-5368Mol Pharm. 2010;7(2):501-9). 

Round 2

Reviewer 3 Report

Accept in present form

Author Response

Thank for your comments